# Seasonal Investigation of Ultrafine Particle Organic Composition in an Eastern Amazonian Rainforest

Adam E. Thomas[1], Hayley S. Glicker[1], Alex B. Guenther[2], Roger Seco[3], Oscar Vega Bustillos[4], Julio Tota[5], Rodrigo A. F. Souza[6], and James N. Smith[1]

[1]Department of Chemistry, University of California, Irvine, USA
[2]Department of Earth System Science, University of California, Irvine
[3]Institute of Environmental Assessment and Water Research (IDAEA-CSIC), Barcelona, Catalonia, Spain
[4]Instituto de Pesquisas Energéticas e Nucleares, Cidade Universitaria, São Paulo, Brazil
[5]Universidade Federal do Oeste do Pará, Santarém, Brazil
[6]Escola Superior de Tecnologia, Universidade do Estado do Amazonas, Manaus, Brazil

**Correspondence:** James N. Smith (jimsmith@uci.edu)

## 1 Abstract

Reports on the composition of ultrafine (<100 nm in diameter) particles in the Amazon are scarce, due in part to the fact that new particle formation has rarely been observed near ground level. Ultrafine particles near the surface have nevertheless been observed, leaving open questions regarding the sources and chemistry of their formation and growth, particularly as these vary
across seasons. Here we present measurements on the composition of ultrafine particles collected in the Tapajós National Forest (2.857° S, 54.959° W) during three different seasonal periods: 10 - 30 September 2016 (SEP), 18 November - 23 December 2016 (DEC), and 22 May - 21 June 2017 (JUN). Size-selected (5 - 70 nm) particles were collected daily (22 h) using an offline sampler. Samples collected during the three time periods were compiled and analyzed using liquid chromatography coupled to Orbitrap high resolution mass spectrometry. Our findings suggest a sustained influence of isoprene organosulfate chemistry on
ultrafine particles in the different periods. We present chemical evidence that biological spore fragmentation impacted ultrafine particle composition during the late wet season (JUN), while chemical markers for biomass burning and secondary chemistry peaked during the dry season (SEP, DEC). Higher oxidation states and degrees of unsaturation were observed for organics in the dry season (SEP, DEC), suggesting greater extents of aerosol aging. Finally, applying a volatility parameterization to the observed compounds suggests organic sulfur species are likely key drivers of new particle growth in the region due to their low
volatility compared to other species.

## 2 Introduction

Ultrafine (<100 nm in diameter) aerosol particles (UAPs) are ubiquitous in the atmosphere, representing the largest fraction of ambient aerosol loading by number (Seinfeld and Pandis, 2006). Inhalation of these particles are linked to adverse health effects in many different parts of the human body, as they are small enough to infiltrate deep into lung tissue, enter the bloodstream,

and cross the blood-brain barrier (Calderón-Garcidueñas and Ayala, 2022; Li et al., 2021; Nassan et al., 2021; Allen et al., 2017). Ultrafine particles can form in the atmosphere from the condensation of low volatility gases in a process known as new particle formation (NPF), considered to be the dominant emission pathway of new particles in the atmosphere (Dunne et al., 2016). Though too small themselves to interact with sunlight at most wavelengths, newly formed particles serve as a significant source of larger particles that can scatter or absorb incident solar radiation, directly impacting climate. In addition, particles formed by NPF are estimated to contribute up to 50% of global cloud condensation nuclei (CCN), particles that can uptake water and serve as seeds for cloud droplets (Gordon et al., 2017; Westervelt et al., 2013; Merikanto et al., 2009; Wang and Penner, 2009). Another study has suggested that UAPs themselves may directly impact cloud formation and drive precipitation over the Amazon basin and other pristine areas by increasing the convective intensity of deep convective clouds (Fan et al., 2018), though this is still an area of active research (Varble et al., 2023). Critical to understanding UAP formation as well as their impacts on human health and climate is information on their distribution and chemical composition as observed in different environments.

Although known for its pristine air quality (Andreae et al., 2004), the Amazon basin nevertheless serves as an aerosol source of global significance thanks in part to its large area and abundant biogenic volatile organic compound (BVOC) emissions (Guenther et al., 2012), directly impacting the global CCN budget as well as helping maintain stratospheric aerosol levels (Williamson et al., 2019; Weigel et al., 2011; Brock et al., 1995). The properties of aerosols formed exhibit a seasonal variability, driven primarily by monthly differences in deposition (i.e., rainfall) and gas emissions (Alves et al., 2016). Despite this significance, the chemistry and dynamics of particle formation and growth over the Amazon is not completely understood. NPF is rarely observed near the ground (Martin et al., 2010), in contrast to remote forested locations in other regions of the world where NPF is observed more frequently (e.g., Andreae et al., 2022; Dada et al., 2018; Dal Maso et al., 2008). Nevertheless, periodic "bursts" of UAPs where little to no growth is apparent have been observed at ground level. For example, a multi-year statistical analysis of size distribution data collected at the Cuieiras forest reservation in central Amazonia revealed that nearly one-third of the observation days featured undefined bursts of UAPs that did not appear to grow from local ground-level NPF events (Rizzo et al., 2018). Although formation is rarely seen near the surface, studies have observed intense NPF activity in the Amazon free troposphere (Andreae et al., 2018; Fan et al., 2018; Wang et al., 2016), where it is thought gaseous precursors are brought by convective updrafts and nucleation occurs due to low temperature and particle surface area conditions (Zhao et al., 2020; Williamson et al., 2019; Weigel et al., 2011). Freshly nucleated particles, in turn, can be brought to the boundary layer by convective downdrafts and grow to larger sizes by coagulation and condensation of BVOC oxidation products (Andreae et al., 2018; Wang et al., 2016; Merikanto et al., 2009; Ekman et al., 2008). A recent study demonstrated that nucleation of organic vapors could serve as the primary formation pathway for these particles (Zhao et al., 2020), but the molecular identities of the species involved and their seasonal variability remain largely unknown. Direct observations of UAP composition could help elucidate these formation and growth pathways and bridge the gap between what is known of the gas emissions and the composition of larger aerosol particles.

Secondary products of BVOCs seem to contribute substantially to aerosol particle mass in the Amazon basin (Martin et al., 2010). Near the surface, it is thought UAP formation is primarily driven by nucleation of sulfuric acid due its extremely low

volatility, after which new particle growth can occur by reactive uptake and condensation of other low volatility species such as oxygenated organic molecules (Ehn et al., 2014; Sipilä et al., 2010). Isoprene, the dominant VOC emitted by regional vegetation (Guenther et al., 2012), has been estimated to contribute substantially to aerosol particle mass principally via the condensation of products formed from reactions with OH radical (Isaacman-VanWertz et al., 2016; Chen et al., 2015). One such product, an organosulfate formed from the heterogeneous uptake of isoprene epoxydiol (IEPOX) onto sulfuric acid seed particles, has been observed in central Amazonia across different seasons, accounting for up to nearly 90% of organic sulfate mass observed in low-NOx conditions (Glasius et al., 2018). Secondary products of monoterpenes have also been observed in Amazon particles, such as 3-methyl-1,2,3-butanetricarboxylic acid (MBTCA) and pinic acid (Leppla et al., 2023; Kourtchev et al., 2016; Kubátová et al., 2000), though perhaps at significantly lower mass concentrations compared to dominant isoprene markers such as IEPOX organosulfate, as suggested by recent findings in $PM_1$ particles in Central Amazonia (Glasius et al., 2018). Due to the lack of particle size resolution in these measurements, particularly in the ultrafine range, it is difficult to comment on the extent to which these species are relevant to particle formation or growth in the region. Glicker et al. (2019) indirectly observed IEPOX in UAPs in central Amazonia during the Observations and Modeling of the Green Ocean Amazon (GoAmazon2014/5) experiment using a thermal desorption chemical ionization mass spectrometer. The dominant positive ion observed throughout the measurement period was tentatively identified as an IEPOX thermal decomposition product (3-methylfuran, *m/z* 83) (Allan et al., 2014), implicating the role isoprene oxidation has on new particle growth in the region. Glicker et al. (2019) also observed significant changes in UAP composition between background or "pristine" periods and periods of anthropogenic emission influence from the city of Manaus (located 70 km west from site). Distinct compositional differences between anthropogenic and background periods were also observed in larger particles ($PM_1$) using an aerosol mass spectrometer at the same site during a different time of the year (de Sá et al., 2019), speaking to the sensitivity of particle growth pathways to human activity in the Amazon. Besides urban emissions, biomass burning has been shown to have a significant impact on particle composition, particularly in months with less rainfall (e.g., de Sá et al., 2019; Kourtchev et al., 2016). Encroaching influence from deforestation, wildfires, and drought are projected to increase in the coming decades (Flores et al., 2024), causing disturbances in the Amazon ecosystem and potentially intensifying regional climate change. Evidence already exists that the distribution of BVOCs in the Amazon are in a state of transition in response to human activity (Yáñez-Serrano et al., 2020). Gaining molecular insight into the species responsible for regional particle formation is critical to efforts that will seek to model how these processes could change and thus impact global climate systems in the future.

Here, we present tree-canopy level observations of UAP molecular composition at a site located within the Tapajós National Forest of eastern Amazonia. To gain insight into the seasonal differences in the sources contributing to particle formation and growth in the region, UAPs were collected during the months September and December of 2016 (dry season) and June of 2017 (late wet season). With an annual rainfall of 1920 mm (Saleska et al., 2003), the Tapajós forest is comparatively drier than other extensive Brazilian wet forests, and has therefore been thought of as a predictive model forest for the larger Amazon ecosystem in future climate change scenarios (Cox et al., 2000).

## 3 Methods

### 3.1 Site description

Measurements were performed at the Santarém-Km67-Primary Forest (BR-Sa1) tower site in the Tapajós National Forest (2.857° S, 54.959° W), located ~7 km west of kilometer 67 of the Santarém-Cuiabá highway and about 50 km south of Santarém, Pará, Brazil. This site is part of the AmeriFlux and Large-scale Biosphere-Atmosphere Experiment in Amazonia (LBA) networks. Tapajós National Forest is comprised primarily of old-growth evergreen species (Saleska, 2019) and experiences a wet season extending from January to June, and a dry season from mid-July through December (da Rocha et al., 2009; Saleska
et al., 2003).

### 3.2 Ultrafine particle collection

Aerosol particles were sampled near the top of the tree canopy by connecting ground-level instruments to a 30 m long inlet constructed of 3/8" copper tubing and sampling at a flow of 13.5 L/min (Fig. S1 in the Supplement). UAPs (5 - 70 nm in mobility diameter) were impacted onto pre-baked aluminum foil substrates using a sequential spot sampler (Aerosol Devices
Inc.) (Fernandez et al., 2014) after being size-selected with a nano differential mobility analyzer (nano-DMA; TSI, model 3085). Particles were sampled into the analyzer at a flow of 3.2 L/min. It should be noted that particles with diameters >100 nm could be sampled when using a nano-DMA for size selection given what is known regarding the charge state distribution of aerosol particles after a Po-210 neutralizer (Liu et al., 1986), though this effect should be reduced by using a selected upper mobility diameter limit <100 nm such as 70 nm used here. Foil substrates were placed inside wells of a 33-well rotating
sample disk, which collected for a period of 22 h per well (midnight to 22:00 local time, each day) followed by a 2 h blank collection in a separate well (22:00 to midnight, each day). The blank consisted of particle-free air sampled by setting the voltage of the nano-DMA to zero, which efficiently removes all particles from the sample stream. Number-size distributions of the collected particles were measured in parallel by counting selected UAPs using a mixing condensation particle counter (Brechtel Manufacturing Inc., model 1720). Periodic size distribution measurements spanning from September 2016 - June
2017 at the Km67 site are shown in Fig. S2. UAPs were collected for compositional analysis during three different seasonal periods: 10 - 30 September 2016 (sample hereafter referred to as SEP), 18 November - 23 December 2016 (DEC), and 22 May - 21 June 2017 (JUN). After collection, sample plates were wrapped in clean aluminum foil, sealed with Teflon tape, and stored at -4 °C until analysis.

### 3.3 LC/MS analysis

Sample foils from each measurement period were pooled together into a vial (1 month/vial) for extraction in 600 $\mu$L of acetonitrile (Sigma-Aldrich, HPLC grade) for 5 h at room temperature (21 °C). Blank collection foils were pooled and extracted in the same manner and number to serve as background signal. Extracts were then transferred into clean vials and evaporated under a gentle stream of nitrogen to a final volume of 50 $\mu$L.

An LC/MS methodology was adopted in the present study to ensure the acquisition of usable spectra with very low sample volume available as well as to reduce artifact-inducing effects of ion suppression that has been observed in the analysis of complex organic aerosols with direct infusion (Kourtchev et al., 2020). Samples were analyzed with an ultrahigh performance liquid chromatograph (Vanquish UHPLC, Thermo Fisher) coupled to an Orbitrap mass spectrometer (Q Exactive Plus, Thermo Fisher), which has a mass resolving power of $\sim 10^5$ at $m/z$ 200 and is equipped with a heated electrospray ionization inlet (HESI). A Luna 1.6 $\mu$m Omega Polar C18 (150 $\times$ 2.1 mm) (Phenomenex) heated at 30 °C was used as the stationary phase, while an eluent system consisting of (A) 0.1% (v/v) formic acid in water (HPLC grade, Sigma Aldrich) and (B) 0.1% formic acid in acetonitrile (HPLC grade, Sigma Aldrich) was used as the mobile phase. A sample volume of 10 $\mu$L was injected at a flow rate of 300 $\mu$L/min and subjected to the following 22 min gradient elution: 3.0 min hold at 95% eluent A, 11 min ramp to 95% eluent B, 2.0 min hold at 95% eluent B, 6.0 min hold at 95% eluent A. The HESI inlet was set to a potential of 2.5 kV and a capillary temperature of 320 °C, with inlet flows of 50, 10, and 1 for sheath, auxiliary (heated to 300 °C), and sweep gases, respectively. A full MS with a data-dependent MS2 acquisition was used for the present study where ions were scanned in the negative ion mode from 100 - 1500 amu (AGC target $10^6$, maximum IT 100 ms), while the top 3 most abundant ions from each scan were selected with an isolation window of 0.4 $m/z$ for MS/MS analysis by higher-energy collisional dissociation (Olsen et al., 2007) (AGC target $5 \times 10^4$, maximum IT 100 ms, NCE levels stepped from 10, 30, and 50). Xcalibur software (version 3.1, Thermo Fisher) was used for data acquisition.

LC/MS data was processed following a methodology developed by Nizkorodov et al. (2011) that has been used in the analysis of organic aerosol samples previously (e.g., Baboomian et al., 2022; Maclean et al., 2021). Briefly, Freestyle software (version 1.6, Thermo Fisher) was used to view and integrate ion chromatograms in the region where analyte signal was observed (1 - 10 min). Decon2LS software (Jaitly et al., 2009) was then used to extract peak positions and intensities. Peak signal from blank extract sample was then subtracted from that in the sample extract. Due to uncertainty regarding whether the air sampled during the blank collections was representative of the air sampled during the particle collections, the background was treated as simply whatever chemical contaminants were on the foil substrates themselves, that is, whatever peaks that should be in the blank extract at around the same level of intensity as the sample. Home-built LabVIEW programs (National Instruments Corp.) were used to remove peaks containing $^{13}$C and $^{34}$S isotopes and assign analyte peaks with the following formula constraints: $C_{1-40}H_{2-80}O_{1-35}N_{0-3}S_{0-2}$, H/C ratios of 0.2 - 2.25, O/C ratios of 0 - 4, double bond equivalents (DBE) of -0.5 - 16.5. Only peaks with $m/z$ 100 - 700 were considered for assignment due to low analyte abundance above this range.

### 3.4 Volatility prediction

Compound volatility distributions were predicted using the assigned molecular formulae from the HRMS data following Li et al. (2016). Volatilities are reported as the saturation concentrations of the pure organic vapors ($C_0$) predicted using the parameterization:

$$\log_{10} C_0 = (n_C^0 - n_C)b_C - n_O b_O - 2\frac{n_C n_O}{n_C + n_O}b_{CO} - n_N b_N - n_S b_S \tag{1}$$

where $n_C$, $n_O$, $n_N$, and $n_S$ refers to the number of carbon, oxygen, nitrogen, and sulfur atoms, respectively. Values used for the parametric coefficients $n_C^0$, $b_C$, $b_O$, $b_N$, and $b_S$ were determined following those provided for the CHO, CHON, and CHOS compound classes in Li et al. (2016).

### 3.5 Meteorology and complementary datasets

Six-hour back-trajectory frequency simulations were calculated over the three seasonal periods discussed in this study using the NOAA Hybrid Single-Particle Lagrangian Integrated Trajectory (HYSPLIT) transport model (https://www.ready.noaa.gov/HYSPLIT.php, last accessed March, 2024) with GDAS 1° meteorology (Rolph et al., 2017; Stein et al., 2015). Fire occurrence data was visualized using the Fire Information for Resource Management System (FIRMS) from NASA (https://www.earthdata.nasa.gov/learn/find-data/near-real-time/firms, last accessed May, 2024), which compiles thermal anomaly data

obtained from the Moderate Resolution Imaging Spectroradiometers (MODIS) aboard the Terra and Aqua satellites (data disclaimer link: https://www.earthdata.nasa.gov/learn/find-data/near-real-time/citation#ed-lance-disclaimer) (Giglio et al., 2018). Meterological ERA5 reanalysis data from the European Centre for Medium-Range Weather Forecasts (Hersbach et al., 2023) was obtained from their Climate Data Store (https://cds.climate.copernicus.eu, last accessed June 2024) to receive parameters of surface temperature, dew point temperature, precipitation, solar radiation at surface, cloud cover, total atmospheric column

water, boundary layer and cloud base height, and leaf area index at the measurement site (2.9° S, -55° W) with 0.25° resolution during each collection period.

### 4   Results and discussion

Fig. 1 shows the average mass spectra obtained for the JUN, SEP, and DEC samples. Only assigned species are shown, corresponding to 77%, 92%, and 76% of the total ion intensity attributed to the the JUN, SEP, and DEC samples, respectively.

Ion peaks are colored based on their assigned molecular formulae, categorized into four compound classes: CHO (green), CHON (blue), CHOS (red), and CHONS (purple). Also included are the ion intensity-based distributions of each compound class in the different months. Percent values shown correspond to the fraction of total assigned ion intensity attributed to the sum of species from each of the four compound classes. The CHO class (species only containing carbon, hydrogen, and oxygen atoms) composed the majority of assigned intensity for all three months (52 - 59%). The CHOS class was the second most

abundant category in JUN and SEP, while the CHON class was more dominant in DEC. CHONS compounds composed the smallest class observed, accounting for only 2 - 6% of assigned ion intensity. The single most abundant ion identified in both JUN and SEP was $m/z$ 215.0231, corresponding to a compound with the electrically neutral molecular formula $C_5H_{12}O_7S$. The molecule is tentatively identified as an IEPOX-derived organosulfate species (IEPOX-OS) based on its formula as well as the prominent loss of bisulfate ($HSO_4^-$) in its MS2 fragmentation spectrum (Fig. S3), a distinguishing feature of organosulfate

species (Attygalle et al., 2001). It is noted that the relative peak intensity of this and other organosulfate species detected could be misleading when comparing to other organic ions present, as ESI ionization is known to be particularly sensitive to organosulfate compounds (Surratt et al., 2008). As noted above, IEPOX-OS species have been observed previously in Amazon

aerosol particles (Glasius et al., 2018; Kourtchev et al., 2016), and their persistent presence in ultrafine particles observed here irrespective of season implicates isoprene organosulfate chemistry as a key player in new particle growth in the Amazon year-round. Despite this consistency, the UAPs sampled also exhibited substantial compositional variation, offering evidence for seasonal variability in emission sources and formation pathways.

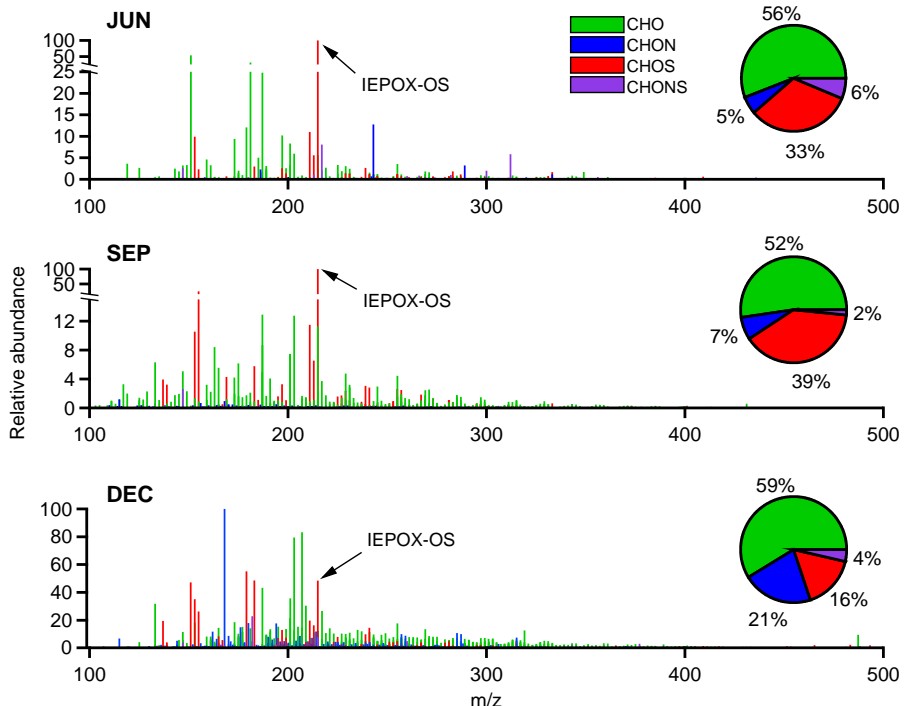

**Figure 1.** Average mass spectra of assigned species in the three months sampled. Color of individual peaks indicates assigned compound class: CHO (green), CHON (blue), CHOS (red), or CHONS (purple). On the right side of each spectrum is overlaid a pie chart displaying the summed ion abundance of each compound class as a fraction of total assigned intensity.

## 4.1 Seasonality of marker ion distributions

A summary of different species that are representative of key sources and processes, herein referred to as "marker ions," have been identified in each month and are shown in Fig. 2A. The species listed represent five different potential sources based on their assigned molecular formula: primary biogenic organic aerosol (PBOA), sesquiterpene-derived secondary organic aerosol (SQT SOA), monoterpene-derived secondary organic aerosol (MT SOA), isoprene-derived secondary organic aerosol (isopene SOA), and biomass burning organic aerosol (BBOA). Together, the marker species included account for 56%, 42%, and 17% of the total assigned ion intensity (or 43%, 38%, and 13% when including unassigned intensity) for JUN, SEP, and DEC, respectively. Table S1 shows a full list of markers identified with suggested chemical structures and references. It

should be noted that since ion percents are calculated as an average over analyte retention time, and since most identifications are based solely on the molecular formulae assigned, the ion signal attributed to species with these formulae may include several isomers. Although CHO compounds comprised the largest fraction of ion intensity in all three months sampled, the most abundant species contributing to this compound class varied with season. During the late wet season (JUN), the top two most abundant CHO compounds were *m/z* 151.0612 (with an electrically neutral formula $C_5H_{12}O_5$) and *m/z* 181.0718 ($C_6H_{14}O_6$), tentatively identified as the sugar alcohols arabitol and mannitol, respectively. These two species have previously been used as markers for PBOA (Samaké et al., 2019; Nirmalkar et al., 2015; Bauer et al., 2008), being found in fungal spores where they are used as energy storage products (Lewis and Smith, 1967). Lawler et al. (2020) previously observed these species in UAPs in the southeastern U.S. using thermal desorption chemical ionization mass spectrometry, finding that the concentration of UAPs containing these species spiked during periods of heightened rainfall. That study concluded that the observed UAPs formed from fungal spores, which fragmented upon exposure to rainfall and/or high humidity. Biological spore samples collected from the Amazon have previously been observed to fragment under high humidity conditions, such as those commonly observed during the wet season, resulting in the formation of UAPs (China et al., 2016). High concentrations of airborne fungal spores have also been linked to rainfall events (Huffman et al., 2013), and particles enriched in biogenic salts linked to fungal spores have been observed in the Amazon wet season (Pöhlker et al., 2012; Lawson and Winchester, 1979). To the best of our knowledge, our observations here represent the first chemical evidence for the occurrence of airborne ultrafine fungal fragments in the Amazon rainforest, as a previously unaccounted source of UAPs in the region. The fact that these ions were more abundant in JUN suggest that this process may be particularly important during the wet season, but more measurements are needed to confirm spatiotemporal prevalence. In addition to arabitol and mannitol, another abundant CHO species in JUN was glucose ($C_6H_{12}O_6$, *m/z* 179.0561), a known marker from plant material such as pollen fragments (Samaké et al., 2019).

The most abundant CHO species in SEP likely were derived from monoterpene oxidation. To help corroborate this, the ion chromatograms and tandem mass spectra obtained for some of these species were compared to those of species observed in α-pinene ozonolysis particles. Bulk ozonolysis particles were collected in a set of laboratory experiments using the spot sampler operating at similar conditions to those used in the present study. Details for these experiments are provided in the Supporting Information. The most abundant species observed was $C_8H_{12}O_5$ (*m/z* 187.0607) and is identified as an isomer(s) of hydroxyterpenylic acid, a proposed marker for α-pinene oxidation (Kahnt et al., 2014). Similar fragments are observed when comparing the MS2 for this ion to literature, including the dominant product ion at *m/z* 125.0239 (ion formula $C_6H_5O_3^-$) (Fig S4). This species was also the most abundant monoterpene oxidation marker identified in JUN and DEC, suggesting that this compound may serve as a suitable marker at different times of the year in the Amazon. An unknown species (excluded from Fig. 2A) with the same unit mass (*m/z* 187.0976, $C_9H_{16}O_4$) found in SEP and DEC has a similar retention time to an α-pinene ozonolysis product observed in our experiments (Fig. S5), although it is likely that several isomers are present with different retention times. The second most abundant CHO species in SEP had a molecular formula $C_8H_{12}O_6$ and is identified as 3-methyl-1,2,3-butanetricarboxylic acid (MBTCA), a marker for the multigeneration oxidation of monoterpenes (Szmigielski et al., 2007). Other studies have found MBTCA in particles from the central Amazon, both observing higher signals in the

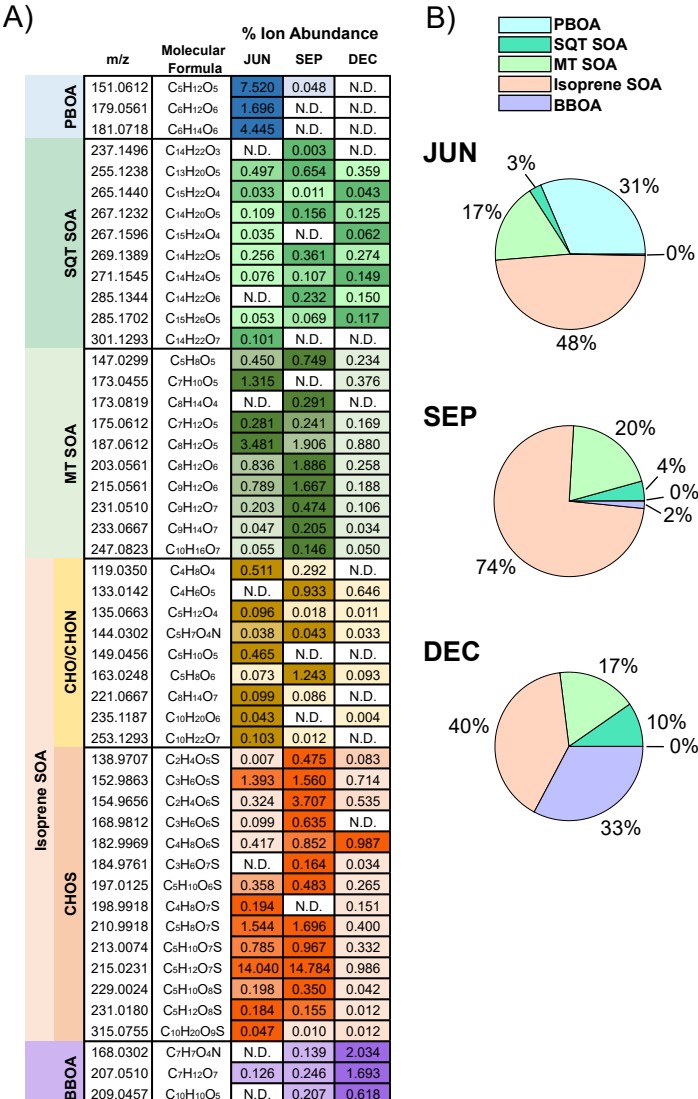

**Figure 2.** Distribution of emission marker species observed in each month. (A) Heatmap of individual ion abundances normalized by total ion abundance (assigned and unassigned) in units of percent. Colors correspond to the different emission source classes: primary biogenic organic aerosol (PBOA, in blue), sesquiterpene-derived secondary organic aerosol (SQT SOA, in dark green), monoterpene-derived secondary organic aerosol (MTSOA, in light green), isoprene-derived secondary organic aerosol (Isoprene SOA, in yellow and orange), and biomass burning organic aerosol (BBOA, in purple). For each ion, darker shades correspond to greater normalized abundance for that species relative to other months. (B) Pie chart showing the contribution of each emission source class as a percent of total marker ion intensity.

dry season (Glasius et al., 2018; Kourtchev et al., 2016) compared to the wet season. This is consistent with the idea that photochemical processing may be more prominent during the dry season when there is likely to be more solar radiation

penetrating to the surface, reaching a maximum around August/September in the nearby city of Santarém (Reis et al., 2022). Corroborating this, ERA5 reanalysis data from the measurement site also shows highest average daytime solar radiation at the surface during SEP (Table S2). It is also worth noting that monoterpene emissions have been observed to peak during the

Amazon dry season (Alves et al., 2022; Yáñez-Serrano et al., 2018), helping explain the overall abundance of these markers during SEP. Comparison of the MS2 spectrum for this ion with that of a product with a similar retention time from the $\alpha$-pinene ozonolysis experiments suggests structural similarities, including evidence for more than one carboxylic acid group (Figs. S6 and S7). It is intriguing that the lower molecular weight MT SOA markers in general show highest abundance in JUN whereas the heavier species starting with $C_8H_{12}O_6$ show highest abundance in SEP, possibly suggesting a seasonal dependence of

reaction branching ratios. Another possible driver of these differences is a seasonal dependence of monoterpene speciation. For instance, Yáñez-Serrano et al. (2018) observed stark seasonal differences in monoterpene distributions in Central Amazonia, observing higher relative abundance of $\alpha$-pinene and limonene in the dry season (August, October) and $\alpha$-terpinene and 3-carene in the late wet season (June). The species $C_5H_8O_5$, identified as 3-hydroxyglutaric acid (Claeys et al., 2007), has also been observed as an oxidation product of isoprene from a non-IEPOX pathway (Liu et al., 2016a; Krechmer et al., 2015),

although it is used as a monoterpene marker here since the IEPOX pathway is likely to dominate during the Amazon wet season (Shrivastava et al., 2019). Relative to JUN and SEP, DEC had a lower contribution of secondary markers from both monoterpenes and isoprene. It should be noted that other compounds of anthropogenic origin such as aromatic hydrocarbons could be driving unaccounted secondary formation as well. Oxidation flow reactor measurements of ambient gases conducted in the Central Amazon previously found that a large fraction of observed SOA formation during the dry season could not

be reconciled using known formation potentials of terpenoids alone (Hodshire et al., 2018; Palm et al., 2018). In addition to monoterpene markers, several sesquiterpene oxidation products were also observed (Fig. 2A), though in lower abundance comparatively. The most abundant product identified had a molecular formula $C_{13}H_{20}O_5$ (*m/z* 255.1238), with a suggested identity of $\beta$-nocaryophyllonic acid, a multigeneration oxidation product of $\beta$-caryophyllene previously identified in Central Amazon PM$_1$ (Yee et al., 2018). In general, observed sesquiterpene products are most abundant in SEP, contributing 1.6%

to total (assigned and unassigned) ion intensity compared to 1.2% in DEC and JUN, consistent with the occurrence of peak abundance in observed monoterpene products. It is notable however that these ion fractions are the most consistent between the three samples compared to all the other emission source classes, suggesting that sesquiterpene chemistry may serve as a year-round source for ultrafine particle growth in the region.

In all three months, isoprene SOA accounted for the largest fraction of the identified marker ion intensity (Fig. 2B). Isoprene

organosulfates were the most dominant isoprene-derived compounds observed for each month, and contributed the most ion abundance to SEP (28% of assigned intensity). Isoprene emissions and the formation of gas phase oxidation products such as IEPOX typically peak in the Amazon dry season due to higher solar intensity and ambient temperatures relative to the wet season (Langford et al., 2022; Rinne et al., 2002), conditions reflected here in a comparison between JUN and SEP (Table S2). Sarkar et al. (2020) observed a strong correlation between isoprene and monoterpene fluxes with temperature and solar

radiation at the same measurement site studied here in the Tapajós National Forest. Riva et al. (2019) found that increasing the concentration ratio of IEPOX to inorganic sulfate drives the formation of organosulfate species. The dominance of these

species observed in SEP suggests that this ratio could be a key driver of organosulfate chemistry in the eastern Amazon, and parallels observations of $PM_1$ composition in the Central Amazon where organic sulfate concentrations peaked in the dry months (Glasius et al., 2018). Seasonal measurements of inorganic sulfate concentrations in UAPs would be needed to probe

the influence of this chemistry further, however. Compared to the other months, SEP also had the lowest average relative humidity (Table S2). Several studies have correlated organosulfate formation with aerosol acidity, and have observed increased uptake of isoprene products such as IEPOX onto acidic particles (Zhang et al., 2018; Liao et al., 2015; Surratt et al., 2007b). Another possible explanation for the dominance of organosulates in SEP could therefore be that there was less water vapor available to dilute the sulfate aerosols needed to form these species, allowing for more efficient reactive uptake to occur.

While still comprising the largest source fraction, the relative contribution of isoprene organosulfates to sample intensity was nevertheless smallest in DEC, most notably IEPOX-OS. Corresponding to the end of the dry season and onset of the wet season, one possible explanation for this is meteorology, given that DEC had the highest average cloud cover (82%) and total column water (52 kg/m$^2$) of the three months studied (Table S2), possibly leading to lower IEPOX formation and uptake for the reasons discussed above. Another possible factor is higher regional levels of NOx due to wildfire activity, leading to the suppression

of the IEPOX pathway by NO (Paulot et al., 2009). Although isoprene is generally expected to eventually form IEPOX via the $HO_2$ oxidant pathway under pristine conditions, Liu et al. (2016b) showed evidence that under polluted conditions where [NOy] > 1 ppb, isoprene photooxidation chemistry drastically shifts towards the NO pathway, producing oxidation products such as methyl vinyl ketone instead of the $HO_2$-produced hydroxyhydroperoxides (ISOPOOH). At the same measurement site in the Central Amazon (located 70 km west of the urban center Manaus), Liu et al. (2018) found that OH concentrations, in-

ferred from measured concentrations of isoprene photooxidation products, increased ~250% under similar polluted conditions, indicating that the anthropogenic contribution of NOx has the potential to greatly impact oxidation pathways. Fig. 3(A) shows air mass history analysis using HYSPLIT and panel (B) shows the distribution of fires observed via satellite from the MODIS instrument over the course of the three seasonal periods. Although each period was dominated by easterly winds, the number of fires observed varied substantially, with the most anomalies occurring in DEC. Corroborating satellite data is the abundance

of identified BBOA species in DEC including $C_7H_7O_4N$, $C_7H_{12}O_7$, and $C_{10}H_{10}O_5$, assigned to methyl-nitrocatechol, methyl galacturonate, and a vanillin oxidation marker, respectively (Mabato et al., 2022; Kong et al., 2021; Iinuma et al., 2010). Under the influence of NOx, isoprene reacts with OH radical and $NO/NO_2$ to form methacryloyl peroxynitrate, which then can be oxidized to form the aerosol markers 2-methylgylceric acid and 2-methylgylceric acid organosulfate (2-MG OS) (Nguyen et al., 2015; Surratt et al., 2007b). Compared to IEPOX-OS, 2-MG OS ($C_4H_8O_7S$, *m/z* 198.9918) was observed to be a comparatively

minor species, but it is notable that the ratio of 2-MG OS to IEPOX-OS was an order of magnitude greater in DEC compared to JUN (Fig. 2A), the month experiencing the least influence from regional wildfire activity (Fig. 3B). The only organosulfate species to have a greater relative contribution to ion intensity in DEC compared to the other months was $C_4H_8O_6S$, *m/z* 182.9969. Shalamzari et al. (2013) observed this compound in ambient filters from Hungary and through structural characterization via $MS^n$ analysis hypothesized that the species was formed from the oxidation of methyl vinyl ketone, a first generation

isoprene product formed via ozonolysis or by OH oxidation in the presence of NOx (Liu et al., 2013; Barket et al., 2004; Aschmann and Atkinson, 1994). Besides isoprene oxidation, methyl vinyl ketone can also be emitted from biomass burning of

tropical forest fuels (Karl et al., 2007) . Additionally, two nitrooxy isoprene OS species (Surratt et al., 2008) were identified in the dry season months (SEP, DEC) (Table S1): $C_5H_{11}O_9NS$ and $C_5H_9O_{12}N_2S$. The identification of these species further supports the hypothesis that isoprene chemistry may have been influenced by NOx enrichment in the dry season. In addition to

wildfire activity, it is also likely that traffic emissions from the Santarém-Cuiabá highway, located ∼7 km east of the measurement site, also contribute to local NOx concentrations. Given the dominance of easterly winds during sampling in each period (Fig. 3A), local highway emissions could impact particle chemistry near the site and the larger Tapajós National Forest area potentially year-round, and could serve as a particularly important source of NOx in the forest during the wet season, when fire activity is low. Besides the influence from NOx emissions, it should be noted that biomass burning can also serve as a source

of $SO_2$ and sulfate (Rickly et al., 2022; Ren et al., 2021). In all, these observations suggest anthropogenic emission sources such as biomass burning could be impacting isoprene organosulfate formation processes in Amazonian UAPs, impacts liable to increase in coming decades with further human activity in the region.

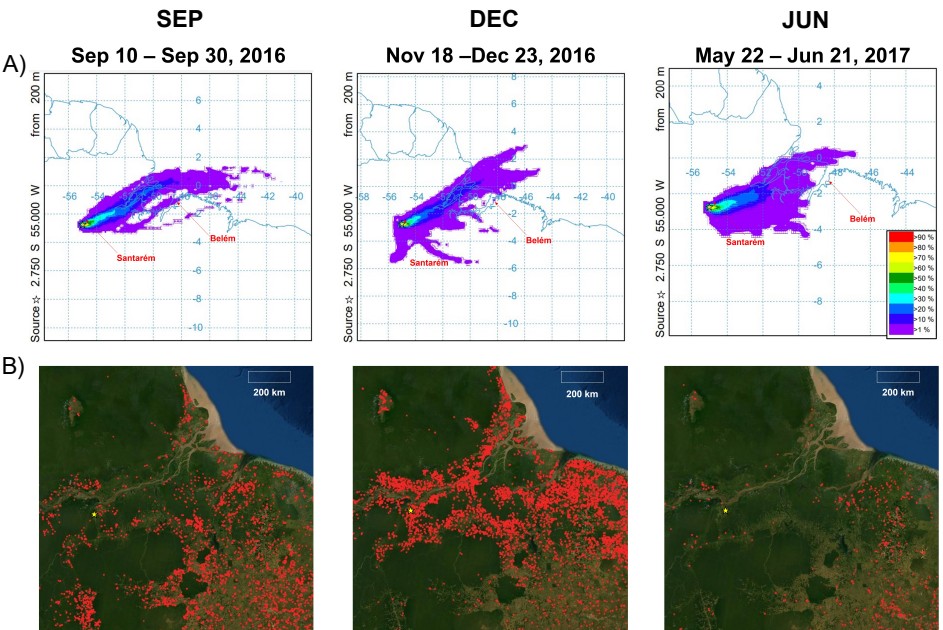

**Figure 3.** (A) Air mass back-trajectory frequencies from HYSPLIT for each seasonal period studied. 48 h back-trajectories were integrated over the time span of each period at a height of 200 m above ground level (near tree canopy). The color scale indicates how frequently air masses passed over a given location, with warmer color indicating higher frequencies. (B) MODIS fire observations from the Aqua and Terra satellites as visualized using FIRMS integrated over each period. Each red dot indicates a thermal anomaly identified as a probable fire occurrence. The yellow star indicates the location of the measurement site in each plot.

Organic sulfur compounds from sources besides isoprene were also identified in all three samples, but most prominently represented in SEP and DEC (Table S1). For instance, the most abundant CHOS species in DEC was $C_6H_{12}O_4S$, *m/z* 179.0385,

and is attributed to the $SO_2$-initiated photooxidation of polycyclic aromatic hydrocarbons, such as those emitted during biomass

burning (Jenkins et al., 1996), and DMSO (Jiang et al., 2022), an oxidation product of dimethylsulfide (DMS), a compound associated with marine emissions (Andreae and Raemdonck, 1983) as well as biomass burning (Meinardi et al., 2003). Other abundant CHOS species in DEC possibly derived from similar sources include $C_4H_8O_4S$ and $C_2H_4O_6S$ (Jiang et al., 2021; Kuang et al., 2016). Monoterpene-derived organosulfate species (MT-OS) were also identified, though in low abundance rel-

ative to isoprene-derived organosulfates, with the sum of MT-OS compounds in each month contributing <0.7% to total ion signal. Interestingly, comparing the different months, the highest number of MT-OS compounds (8) were identified in DEC, including $C_7H_{12}O_7S$, $C_9H_{16}O_6S$, and $C_7H_{12}O_6S$. In addition, two nitrooxy MT-OS species were identified and solely present in DEC: $C_{10}H_{17}O_7NS$ and $C_9H_{15}O_8NS$. These species have previously been proposed to form from either nitrate radical oxidation or photooxidation in the presence of NOx, both in the presence of acidic sulfate particles (Surratt et al., 2008).

Species with molecular formulae of non-organosulfate isoprene markers were also identified in each month (Fig. 2A), including 2-methyltetrol ($C_4H_8O_4$, *m/z* 119.0350) and 2-methyltartaric acid ($C_5H_8O_6$, *m/z* 163.0250) (Jaoui et al., 2019; Claeys et al., 2004). Interestingly, most of the CHO isoprene markers identified were most abundant in JUN. Given that this month experienced the highest average relative humidity, a possible explanation would be that these products were formed under less acidic conditions in particles, hindering the subsequent oxidation by sulfuric acid to form their organosulfate counterparts.

Regarding possible humidity effects, it is also worth noting that 2-methyltetrol, formed from the particle-phase hydrolysis of IEPOX, and its dimer ($C_{10}H_{22}O_7$, *m/z* 253.1293) are most abundant in JUN (Armstrong et al., 2022; Riva et al., 2019; Surratt et al., 2007a). Conversely, the IEPOX-OS dehydration product $C_5H_{10}O_6S$ (*m/z* 197.0125) is most abundant in SEP, the driest month observed. Finally, a marker for the oxidation of isoprene by nitrate radical was also identified ($C_5H_7O_4N$, *m/z* 144.0302) (Rollins et al., 2009). The ion fraction of this species was rather consistent across all three months, suggesting that this process

may serve as a regular (likely nighttime due to reaction with nitrate) source of UAP growth in the region irrespective of season.

## 4.2 Seasonality of aerosol molecular properties

In addition to differences in the marker ions, differences in the molecular properties of all species observed in each month such as the distribution of oxidation states and degrees of unsaturation were also investigated. Fig. 4 depicts the carbon oxidation states and double bond equivalents of the CHO-containing molecules observed in each period. Carbon oxidation state ($OS_C$)

was estimated following Kroll et al. (2011):

$$OS_C \approx 2O/C - H/C \qquad (2)$$

where O/C and H/C refer to the ratio of oxygen to carbon atoms and hydrogen to carbon atoms, respectively. It should be noted that the presence of some functionalities such as peroxides will lead to an overestimation of carbon oxidation state. Nevertheless, this estimate serves as a relative means to compare the extent of oxidation of molecules in each period. Most species

observed had $OS_C$ values between -1 and 1, corresponding to the oxidation range associated with semi- and low volatility oxidized organic aerosol components (Kroll et al., 2011). Compared to JUN, higher oxidation states were observed in SEP and DEC across most carbon numbers. Using measurements of reactive gases, previous studies have shown that OH reactivity is highest during the Amazon dry season (Pfannerstill et al., 2021; Nölscher et al., 2016) coinciding with the seasonal peak in

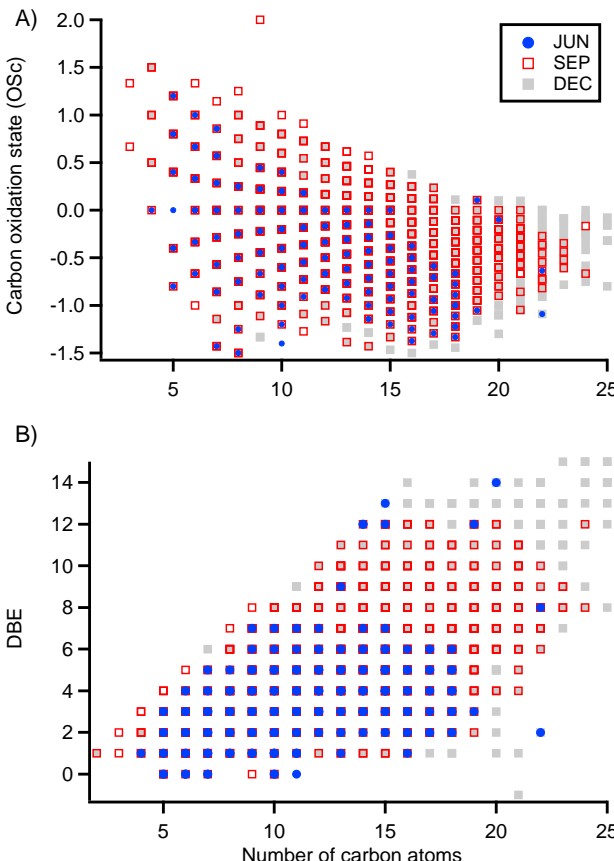

**Figure 4.** Comparison of (A) carbon oxidation state and (B) double bond equivalents of CHO-containing molecules observed in each seasonal period.

BVOC emissions and sunlight. These factors, coupled to the fact that potentially fewer products are lost due to wet deposition, can thus be thought of as optimized for the multigeneration oxidation of organics during the dry season. Higher molecular weight CHO species were also observed in SEP and DEC. Baboomian et al. (2022) found that UV-aged SOA components were on average more oxidized, unsaturated, and heavier than those from fresh aerosol, attributing this to condensed-phase photochemical reactions. In line with this, CHO species observed in JUN were also more saturated compared to SEP and DEC (Fig. 4B). Besides the possible influence of photochemistry, biomass burning organics can also be highly unsaturated, with oxygenated aromatic and nitroaromatic species as known markers (Bianchi et al., 2019; Iinuma et al., 2010).

Fig. 5 depicts the distribution of highly unsaturated CHON molecules (DBE $\geq$ 6) across the periods. CHON was the smallest compound class in JUN by ion abundance, and had the lowest number of observed molecular formulae (65 molecules assigned), most of which were relatively saturated. While more unsaturated species were observed in SEP, the most highly unsaturated CHON species were observed in DEC, possibly due to wildfire influence. Kourtchev et al. (2016) observed a similar trend in

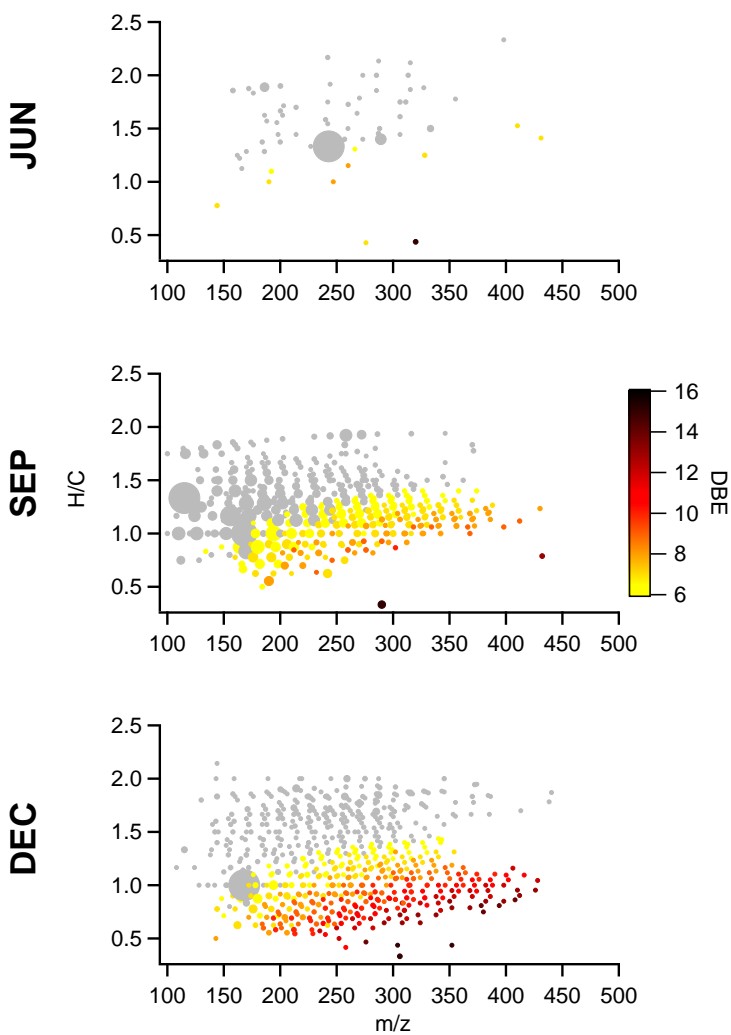

**Figure 5.** Distribution of double bond equivalents (DBE) for CHON-containing molecules observed in each seasonal period. Marker size indicates ion relative abundance. Molecules with DBE < 6 are shown in gray.

bulk organic aerosol collected from Central Amazonia, also suggesting the influence from BBOA aromatics. Fig. S8 shows a similar trend apparent for CHO molecules. Fig. 6 depicts the distribution of CHON species in Van Krevelen space. In addition to being more saturated, CHON molecules observed in JUN were also less oxidized compared to the peak dry season (SEP), suggesting that aging processes such as photochemistry that could be affecting the CHO fraction are likely also influencing nitrogen-containing species in the dry months. A similar trend is observed when comparing JUN to DEC as well (Fig. S9),
though most of the most abundant ions appear to have similar O/C ratios.

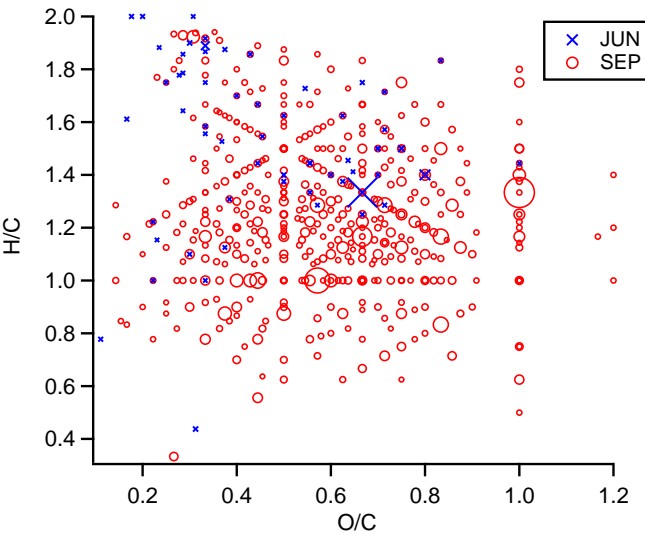

**Figure 6.** Van Krevelen diagram of the CHON molecules observed in JUN (blue circles) and SEP (red squares). Marker size indicates relative ion abundance.

Using a molecular corridor parameterization (Li et al., 2016), the volatilities of UAP constituents were estimated for each compound class (Fig. 7) to investigate relative changes in the species volatility distributions across the different seasons. Most species observed in each period were either classified as semivolatile organic compounds (SVOC), defined as having saturation mass concentrations $C_0$ between 0.3 and 300 $\mu$g/m$^3$, or low volatility organic compounds (LVOC) ($3 \times 10^{-4} < C_0$
$< 0.3$ $\mu$g/m$^3$) (Murphy et al., 2014; Donahue et al., 2011). Although these species are most likely not present at concentrations needed to initiate particle formation at the surface, it should be noted that many species classified as SVOC or LVOC at standard temperature and pressure (298 K, 1 atm) are classified as lower volatility species in the upper atmosphere, where temperatures are much lower. This is consistent with prior observations that Amazon UAPs are largely formed in the free troposphere (Zhao et al., 2020). Fig. S2 shows particle size distributions at the site from September 2016 through June 2017 (Fig. S2A), as
well as the time-averaged distributions for the three seasonal periods (Fig. S2B). No clear NPF events were observed at the surface during any of the periods, though highest UAP number concentrations were detected during the drier months (late June, September - December). Previous studies (Franco et al., 2022; Rizzo et al., 2018) have observed a similar seasonal dependence of particle concentrations in central Amazonia, with the dry season being dominated by accumulation mode ($\sim$100 - 600 in diameter) particles, possibly contributed by regional biomass burning events. The average size distributions bear a striking
resemblance across the different periods, suggesting that any differences in aerosol molecular properties including volatility are not due to the sampled particle size.

Isoprene organosulfates including IEPOX-OS ($C_5H_{12}O_7S$, $C_0 = 0.16$ $\mu$g/m$^3$) dominate the LVOC fraction, particularly in JUN and SEP (Fig. 7). This is consistent with Lopez-Hilfiker et al. (2016), who classified IEPOX SOA components as low volatility empirically by studying thermal desorption characteristics. However, the actual value estimated for $C_0$ here is

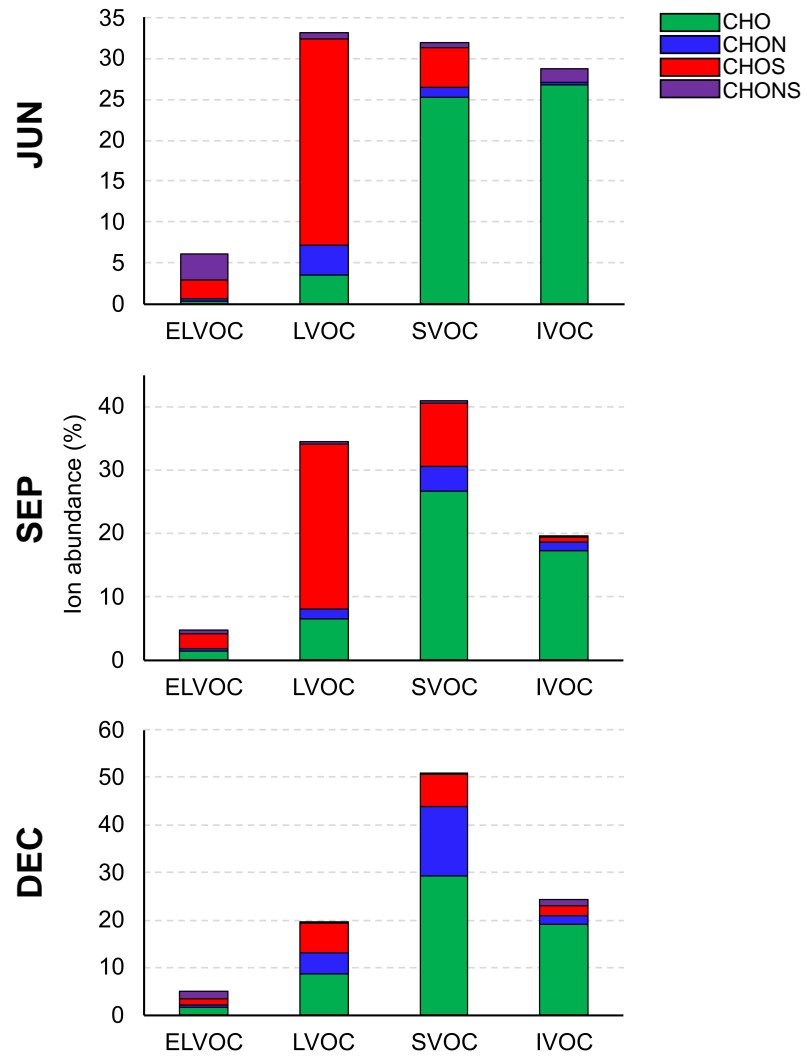

**Figure 7.** Compound class volatility distributions calculated for each seasonal period. Ion abundance (%) is relative to each sample.

several orders of magnitude higher than that estimated by Lopez-Hilfiker et al. (2016). It has been demonstrated previously that the parameterization developed by Li et al. (2016) can significantly overestimate $C_0$ for certain atmospherically relevant compounds including nitrogen-containing species (CHON) (Isaacman-VanWertz and Aumont, 2021), likely due to inherent functional group biases in the training set of molecules used to develop the parameterization. Given that the most abundant CHOS molecule observed here is very likely an organosulfate species (Fig. S3), and that only 17 of the 925 compounds used in the training set by Li et al. (2016) were organosulfates, it is possible a similar bias is observed here. Nevertheless, the low volatility (if not lower) and high relative abundance of isoprene organosulfates observed in this study imply that these species are likely key contributors to UAP growth in the region. In addition to isoprene-derived species, other CHOS molecules likely

contribute significantly as well, as many of the individual species are classified as either LVOC or ELVOC ($C_0 < 3 \times 10^{-4}$) (Fig. S10). Given that easterly winds were dominant during the sampling periods (Fig. 3A), a key source of sulfate is likely oxidation

of dimethylsulfide (DMS) given its aforementioned association with marine environments (Andreae and Raemdonck, 1983), which forms methanesulfonic acid and sulfuric acid as secondary products (von Glasow and Crutzen, 2004). The Amazon ecosystem itself has also been identified as another source of DMS, with emissions from plants and soils shown to be strongly dependant on light and temperature (Jardine et al., 2015). In addition to DMS, regional sulfate production and thus UAP growth is likely highly sensitive to any contributions from $SO_2$ emission sources such as biomass burning and fossil fuel combustion

on land and at sea (Andreae, 2019; Lee et al., 2011). The abundant biomass burning-related species including $C_7H_7O_4N$ ($C_0$ = 296 $\mu g/m^3$) and $C_7H_{12}O_7$ ($C_0$ = 7.6 $\mu g/m^3$) fall into the SVOC fraction, contributing to the dominance of this fraction in DEC. Also contributing to the SVOC fraction are several MT SOA markers such as MBTCA ($C_8H_{12}O_6$, $C_0$ = 94 $\mu g/m^3$) and $C_{9-10}$ monomer species. It is noteworthy that the $C_0$ estimated by Thoma et al. (2022) for MBTCA using the SIMPOL.1 model, which considers the volatility contribution of specific functional groups including carboxylic acids, is four orders of

magnitude lower than the estimate used here, and therefore could be classified as an LVOC. It is reasonable to suspect other species containing carboxylic acid moieties, particularly more than one, may be overestimated as well, causing uncertainty in the volatility distributions of each of the compound classes used here, all of which contain oxygen. CHO monomers with $C_{<8}$ associated with monoterpene and isoprene oxidation are generally classified here as intermediate volatility species (IVOC) ($300 < C_0 < 3 \times 10^6$ $\mu g/m^3$). Though functional group elucidation may prove them to have a lower volatility due to the

suspected abundance of hydroxyl moieties (Donahue et al., 2011), the PBOA sugar markers are classified here as either SVOC (glucose and mannitol) or IVOC (arabitol) species. In general, the molecular volatility distributions taken together with the size distribution data suggest that the secondary UAPs measured here were likely nucleated elsewhere such as over the Atlantic ocean (e.g., sulfuric acid nucleation) or the Amazon free troposphere (e.g., organic vapor nucleation) and what was sampled are mostly aged aerosol components ranging from low to intermediate volatility. UAPs above nucleation sizes (>10 nm in

diameter) are also likely contributed from other sources at the regional level such as biomass burning in the dry season (SEP and DEC) and biological spore emissions in the late wet season (JUN).

## 5 Conclusions

UAPs collected from an eastern Amazon rainforest during three distinct seasonal periods exhibited compositional differences that reflect seasonality in emission sources and aerosol molecular processing. Our observations suggest isoprene oxidation

chemistry is likely a key contributor to UAP composition in the eastern Amazon year-round, and that any regional perturbations in isoprene and/or sulfate emission sources could have a significant impact on UAP formation and growth characteristics in the future. Known markers for biological spore emissions were tentatively identified in UAPs with peak abundance observed in the late wet season sample (JUN). To the best of our knowledge, this finding provides the first chemical evidence for the influence of primary biogenic sources on UAP composition in the Amazon. To better assess this source's relative importance,

future measurements are needed to establish the prevalence of spore emission and fragmentation throughout the wet season and

in different regions of the Amazon. Secondary emission sources related to the oxidation of isoprene and monoterpenes appear to contribute most to UAP composition during the peak dry season (SEP), aligning with the seasonal peak in VOC emissions and solar radiation. In the late dry season (DEC), biomass burning appears to have a substantial impact on UAP composition, as indicated by the abundance of tentatively identified species such as methyl-nitrocatechol and corroborated by MODIS fire

anomaly data. Future emission scenarios in which wildfire and agricultural burning events are more frequent could therefore significantly alter regional UAP concentrations and growth characteristics during this time of year. Inasmuch as the Tapajós rainforest is viewed as a model for the future of the Amazon (Cox et al., 2000), our study provides evidence that this source will likely have increasing influence on UAP emissions in other regions such as Central Amazonia as well.

Reflecting the differences observed in the distribution of suspected marker species, seasonal changes in the properties of all

UAP molecular constituents were also apparent. Compared to those observed in the late wet season, CHO and CHON organics detected in the dry season were more oxidized and unsaturated, suggesting a difference in the extent of aerosol processing such as photochemical aging. Examining the distribution of molecular volatilities reveals that while a strong presence of low volatility species were found in the late wet and peak dry season due to the abundance of isoprene-derived organosulfates, semi-volatile species likely emitted from biomass burning dominate the volatility distribution in the late dry season. Although

the strong presence of low volatility organosulfates suggests that many of the sampled UAPs formed via sulfuric acid nucleation occurring to the east of the site, it is unclear whether semi-volatile species contributed from biomass burning condensed on these seeded particles or were emitted/formed by a separate mechanism. Similarly, it is unclear what fraction of observed semi-volatile BVOC oxidation products contributed to condensational growth of seeded particles near the surface or nucleated new particles in the upper atmosphere. Future studies on the mixing state of Amazon UAPs would help us more quantitatively

understand how the identified emission sources fit in the formation and growth history of new particles in the region. Regardless, it is clear both from the year-round abundance of sulfated species as well as the seasonal abundance of biomass burning markers in UAPs that human activity is likely already playing a role in formation and growth processes, and will indeed have greater influence in a probable future.

*Data availability.*   Data are publicly available and are archived at https://doi.org/10.5061/dryad.k6djh9wgb (Smith, 2024)

*Author contributions.*   JNS and JT secured funding, conceived research concept and made on-site measurements. JNS and HSG retrieved samples and analyzed size distribution data. AET analyzed samples and visualized data under supervision of JNS. AET prepared the manuscript with contributions from all authors.

*Competing interests.*   The authors declare that they have no conflict of interests.

*Acknowledgements.* The authors acknowledge funding from the Brazilian Science Mobility Program (Programa Ciência sem Fronteiras) Special Visiting Researcher Scholarship. RS acknowledges a Ramón y Cajal grant (RYC2020-029216-I) funded by MCIN/AEI/ 10.13039/501100011033 and by "ESF Investing in your future". IDAEA-CSIC is a Severo Ochoa Centre of Research Excellence (MCIN/AEI, Project CEX2018-000794-S). We acknowledge the use of imagery from NASA's Fire Information for Resource Management System (FIRMS) (https://earthdata.nasa.gov/firm part of NASA's Earth Science Data and Information System (ESDIS). JNS acknowledges funding from the US Department of Energy (DE-SC0012704). The authors thank Véronique Perraud, Deanna C. Myers, Michelia Dam, Paulus S. Bauer, Madeline E. Cooke, Jeremy Wakeen, Anna Kapp, Colleen E. Miller, Kristen L. Kramer, and Berenice Rojas for their contributions to discussions regarding this study.

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
