# Peer review of "Seasonal Investigation of Ultrafine Particle Organic Composition in an Eastern Amazonian Rainforest"

_EGUsphere, 2024_

## Referee Comment (RC2)

Accept, minor revisions

The manuscript entitled "Seasonal Investigation of Ultrafine Particle Composition in an Eastern Amazonian Rainforest" by Thomas et al. presents an analysis of the composition of ultrafine particles (<100 nm) in the Tapajós National Forest during three seasonal periods. The authors collected size-selected particles (5-70 nm) and analysed them using high-resolution mass spectrometry. The results show that isoprene organosulfate chemistry consistently influenced ultrafine particles throughout the seasons, while biological spore fragmentation influenced particle composition in the late wet season. Biomass combustion and secondary aerosol chemistry were prominent during the dry season, leading to higher oxidation states in the particles. Organic sulfur species, due to their low volatility, are suggested to be the main drivers of new particle growth in the region.

The findings in this manuscript are certainly a valuable addition to the existing literature and my recommendation is that the manuscript should be published after some minor changes and improvements. From a scientific perspective, the manuscript clearly fits the scope of ACP. It is a thorough and interesting study, relevant for the field of atmospheric research and beyond. From a formal perspective, the quality of the manuscript is high - it is well-written, the figures and tables are clear, and all arguments and aspects are presented clearly.

Below, you will find some general comments that the authors may want to consider:

Page 2, line 33: the term "aerosol reservoirs" seem misleading here – I suggest to look for a more appropriate word.

Page 2, line 38: Please cite some more and ideally more recent studies on the absence of NPF in the Amazon here.

Page 6, line 176f.: Are there any indication for sesquiterpene-derived SOA in the data?

Page 4, line 89: The measurement site is west (= downwind) from the rather heavily used highway BR163. What does this mean for the measurements and data? How would signals from traffic or other human-made emissions along the highway show up in the aerosol sizing data? How can potential fossil fuel combustion be excluded as contamination for the chemical signatures? Further, how representative is this site for UFA processes in the Amazon?

Page 4, line 97: how can it be excluded that multiply charged particles are also collected in the size range from 5 to 70 nm? Could this mix larger-sized particles in the ultrafine size range?

The collection of aerosol particles down to 5 nm can be experimentally challenging. How was the inlet at the tower designed and what does the transmission curve look like for the flow rate and sampling conditions chosen here?

The aerosol number size distributions seem a bit high – I created an overplot of the size distributions of the present study and selected previous Amazonian studies (see figure below). How can the discrepancy in dN/dlogD be interpreted?

[Figure]

Page 15, line 346ff: Cite also Franco et al. ACP, 22, 3469–3492, https://doi.org/10.5194/acp-22-3469-2022, 2022 here.

---

## Author Comment (AC1)

**Responses to Reviewer Comments**
**Seasonal Investigation of Ultrafine Particle Organic Composition in an Eastern Amazonian Rainforest**

A. E. Thomas, H. S. Glicker, A. B. Guenther, R. Seco, O. Vega Bustillos, R. A F. Souza, and J. N. Smith

Department of Chemistry, University of California, Irvine, CA 92697

**Note to editor:** Based on a referee comment, the authors decided to modify the title of the manuscript to the following (same as above): "Seasonal Investigation of Ultrafine Particle Organic Composition in an Eastern Amazonian Rainforest".  This change has been noted in the markup and included in the final version of the manuscript.

Color code for document: Blue – Referee comments, Black – Author responses, Red – Manuscript edits (including LaTeX code)

RC1: Alexander Vogel, 26 Aug 2024

Thomas et al. present a seasonal study on the composition of ultrafine particles in the Eastern Amazon. Strictly speaking, on the composition of the organic fraction in ultrafine particles, as no information on inorganic ions (sulfate, nitrate, ammonia) or metals is provided. In this respect, the title should contain that information, especially because in the light of publications of the Smith group, one might expect the measurements with the TD-CIMS including inorganic ions. In this study, they sampled particles with a combination of a nano-DMA and a sequential spot sampler for offline analysis. Samples from three different months were merged and analysed by liquid chromatography / high-resolution mass spectrometry with negative-mode electrospray ionization. The samples from June represent the late wet season, September the dry season and December the late dry season with strongest biomass-burning activities. They found in all three samples strong contributions from isoprene-derived organosulfates. For the late wet season, they also detected compounds that are likely emerging from fragments of biological spores. The December sample showed strongest aromaticity (degree of unsaturation) and highest average carbon oxidation state. Overall, the manuscript presents substantial novel data, the results are discussed in a balanced and appropriate way with plenty of relevant references. The presentation quality is good to excellent – I would only suggest to adapt to the colour logic of the AMS community (Figure 1 and 7 with CHO as green; CHON and CHOS

are already good with blue and red). Apart from this minor thing, I have one important comment on the volatility classification, as described in the next paragraph. I think this can be resolved, so overall, there is one major issue to resolve and a few minor clarification or changes necessary.

We thank the reviewer for their insightful feedback and comments, which has helped improve the manuscript. We agree that this is certainly an organic composition paper and have modified the title to the following based on the above recommendation: "Seasonal Investigation of Ultrafine Particle *Organic* Composition in an Eastern Amazonian Rainforest". We have changed the color logic as suggested for Figures 1 and 7 (CHO black --> green, CHONS green --> purple). Responses to other comments and manuscript edits are addressed below. Line numbers correspond to the final edited version of the manuscript.

Volatility: the authors use the volatility classification of Li et al. (2016) – the molecular corridor approach. This is justifiable, because this approach is the only volatility classification that allows to calculate C* for compounds that include all three relevant heteroatoms: oxygen, nitrogen, and sulphur. However, the Li et al. parametrisation tends to overestimate C* up to a few orders of magnitude, as pointed out by Isaacman-VanWertz and Aumont (2021) and Thoma et al. (2022). In line 363, the authors classify the tricarboxylic acid MBTCA as semi-volatile based on Li et al. (2016). Thoma et al. (2022) argue instead (on the same molecule) that with using the SIMPOL.1 model, which specifically considers the carboxylic acid functionality, MBTCA is a low-volatile compound. Almost four orders of magnitude lower C* between the two different approaches – with this bias one might come easily to wrong conclusions about new particle formation and growth. In my opinion, the SIMPOL.1 model is more trustworthy as it specifically accounts for functional groups. If the sulphur-containing species that were used in the Li et al. parametrisation are no organosulfates, this might also cause systematic uncertainties. However, due to the polarity of organosulfates, the conclusion that these compounds are low-volatile and contribute to particle mass appears plausible. I think this issue can be resolved with an appropriate discussion of the uncertainty regarding volatility prediction.

We thank the reviewer for highlighting this critical point of uncertainty in our volatility estimation. The fact that each of the compound class bins used here (CHO, CHON, CHOS, CHONS) include oxygen and therefore the possibility of containing species with carboxylic acid moieties makes the reference regarding the reduced volatility of MBTCA particularly relevant. We also agree that the main conclusion we draw from our calculation regarding the low relative volatility of CHOS molecules compared to other classes remains plausible, given that the abundance of organosulfates likely causes an overestimation of C* rather

than an underestimation.   We nevertheless include a statement discussing this uncertainty when presenting our estimate for IEPOX-OS (line 384):

However, the actual value estimated for $C_0$ here is several orders of magnitude higher than that estimated by \citet{Lopez_Hilfiker2016}.  It has been demonstrated previously that the parameterization developed by \citet{Li2016} can significantly overestimate $C_0$ for certain atmospherically relevant compounds including nitrogen-containing species (CHON) \citep{Isaacman_VanWertz2021}, likely due to inherent functional group biases in the training set of molecules used to develop the parameterization. Given that the most abundant CHOS molecule observed here is very likely an organosulfate species (Fig. S3), and that only 17 of the 925 compounds used in the training set by \citet{Li2016} were organosulfates, it is possible a similar bias is observed here.  Nevertheless, the low volatility (if not lower) and high relative abundance of isoprene organosulfates observed in this study imply that these species are likely key contributors to UAP growth in the region.

We also included a statement addressing the uncertainty in the estimation for MBTCA and other potential carboxylic acid-containing molecules (line 403):

It is noteworthy that the $C_0$ estimated by \citet{Thoma2022} for MBTCA using the SIMPOL.1 model, which considers the volatility contribution of specific functional groups including carboxylic acids, is four orders of magnitude lower than the estimate used here, and therefore could be classified as an LVOC.  It is reasonable to suspect other species containing carboxylic acid moieties, particularly more than one, may be overestimated as well, causing uncertainty in the volatility distributions of each of the compound classes used here, all of which contain oxygen.

We finally clarify the purpose of estimating the volatility when first presenting results from our calculation to avoid potential misunderstandings regarding the absolute values presented, as we only wish to draw conclusions from relative differences in compound class and season studied (line 366):

Using a molecular corridor parameterization \citep{Li2016}, the volatilities of UAP constituents were estimated for each compound class (Fig.~\ref{volatility}) to investigate relative changes in the species volatility distributions across the different seasons.

Further comments:

1. line 46-48: what happens to organic aerosol partitioning during convective downdrafts? BVOCs that were nucleating/condensing in the upper troposphere at -50°C might re-evaporate when reaching the surface layer?

This is an interesting point.  It is true that evaporation likely occurs when particles reach the surface layer. There may be competing effects to consider for the case of convective precipitation. This could create optimal conditions for small particle growth in a downdraft, as there may be lower background particle surface area that combines with cooler temperatures from water evaporation.  We choose not to comment on this phenomenon in the manuscript, as it is already the focus of research for other studies in the Amazon (e.g., Wang, J., Krejci, R., Giangrande, S. et al. Amazon boundary layer aerosol concentration sustained by vertical transport during rainfall. Nature 539, 416–419 (2016). https://doi.org/10.1038/nature19819, *cited in text*).

2. l. 61-62: how does MBTCA and pinic acid relate quantitatively to isoprene SOA markers?

To the best of our knowledge the relative abundance of products of monoterpene oxidation compared to isoprene remains unclear for Amazon ultrafine particles.  Glasius et al. (2018) did present mass concentrations for MBTCA and pinonic acid in PM1 from particles in the Central Amazon and compared these to IEPOX-OS, finding the isoprene organosulfate to be an order of magnitude or greater concentration across both seasonal periods studied.  This finding has been incorporated into the introduction to add to the discussion of what is known regarding these markers in larger particles (line 63):

Secondary products of monoterpenes have also been observed in Amazon particles, such as 3-methyl-1,2,3-butanetricarboxylic acid (MBTCA) and pinic acid \citep{Leppla2023, Kourtchev2016, Kubatova2000}, though perhaps at significantly lower mass concentrations compared to dominant isoprene markers such as IEPOX organosulfate, as suggested by recent findings in PM$_1$ particles in Central Amazonia \citep{Glasius2018}.

3. l. 95: 30 meters is a very long inlet. This is problematic especially for ultrafine particles. Were size-dependent inlet line losses quantified? If not, they should be estimated and it should be critically mentioned that the overall 5-70 nm range is systematically biased towards larger particles. On the same issue: what about multiple-charged particles and their contribution? Overall, if a differential mobility analyser was used for size selection, the associated uncertainties should be discussed in the methods section.

Considering the dimensions of the 3/8" copper tubing used here as well as the 13.5 L/min sampling flow, size-resolved inlet losses were calculated and are shown below as a particle

penetration efficiency curve.  Also plotted are the particle-volume size distributions for the three sampling periods, plotted using the color scheme throughout the paper (blue = JUN, red = SEP, black = DEC).

[Figure]

As we can see, particle loss is considerable below 10 nm (up to 40%) but in terms of their contribution to particle volume sampled, these losses are negligible.  Most of the particle volume collected was in fact from the highest size bins (upper limit 70 nm).    The penetration efficiency curve was added to the supplementary information (Fig. S1).

We added some additional details concerning our inlet in the main text (line 97):

 Aerosol particles were sampled near the top of the tree canopy by connecting ground-level instruments to a 30~m long inlet constructed of 3/8" copper tubing and sampling at a flow of 13.5 L/min (Fig. S1 in the Supplement).

In the same section, we also noted the flow rate used by our DMA-spot sampler system (3.2 L/min) (line101):

Particles were sampled into the analyzer at a flow of 3.2 L/min.

The effect of multiple charges is certainly a factor, but fortunately most of the multiply charged species collected should still have been in the ultrafine range by volume. Considering particles with the upper electrical mobility diameter limit of 70 nm as an example, and assuming (1) a Boltzmann distribution of charges as one would expect after a Po-210 bipolar neutralizer and (2) for simplicity the number concentration of particles larger than 70 nm is equal to the concentration at that size (which is reasonable considering what we know about size distributions in the Amazon), we can estimate that 84% of the particles collected are singly charged, 14% are doubly charged, and 2% are triply charged, and so on.  Upon calculating a volume-weighted mean diameter collected after considering these first three charge states, we find that in this scenario the average particle size collected has a diameter of 79 nm, well within the ultrafine range.

A note regarding the uncertainty in using a nano-dma for particle size selection was added to the main text (line 101):

It should be noted that particles with diameters $>$100 nm could be sampled when using a nano-DMA for size selection given what is known regarding the charge state distribution of aerosol particles after a Po-210 neutralizer \citep{Liu1986}, though this effect should be reduced by using a selected upper mobility diameter limit $<$100 nm such as 70 nm used here.

4. l. 99: shouldn´t the blank also be sampled for 24 hours? I guess the two hours were chosen to get daily blanks; the shorter sampling duration might cause blanks appearing too low? Also in this section on the sampling, I miss the information on the sampling times. Midnight to midnight?

Two important points that should be clarified.  First, blanks were collected every day from 22:00 to 00:00 (local time).  Second, regarding how to consider these blank collections as "background" and subtract them for the analysis, we thought it best not to assume that what was sampled during the 2-hour blank measurement was representative of each 22-hr collection.  Therefore, we decided to use as our background simply the foil substrates themselves, that is whatever chemical contaminants that should have been on each foil at around the same level.  To account for this, we extracted the same number of blank collection foils as actual collection foils for each plate (each month) and for the analysis all peaks present in this blank extract were subtracted from those in the sample extract.

We have clarified the blank sampling time in the methods (line 104):

Foil substrates were placed inside wells of a 33-well rotating sample disk, which collected for a period of 22 h per well (midnight to 22:00 local time, each day) followed by a 2 h blank collection in a separate well (22:00 to midnight, each day).

We also clarify that the foils extracted from the collections and blanks were equal (line (116):

Blank collection foils were pooled and extracted in the same manner and number to serve as background signal.

We also include an explanation for how the background was treated for the sake of the analysis (line 138):

Peak signal from blank extract sample was then subtracted from that in the sample extract. Due to uncertainty regarding whether the air sampled during the blank collections was representative of the air sampled during the particle collections, the background was treated as simply whatever chemical contaminants were on the foil substrates themselves,

that is, whatever peaks that should be in the blank extract at around the same level of intensity as the sample.

5. l. 116: m/z in italics (check the whole manuscript).

This error was amended for the whole manuscript.

6. l. 127: This sounds as if the raw data were investigated manually for peaks? I strongly suggest free non-target analysis software such as mzmine. With this software the risk of missing peaks is much reduced and files can be processed in batch mode. Also it allows to use data-bases for compound identification, and possibly identify oxidation products of VOCs mentioned in line 225-228.

We will likely be using something like this for future work, thank you for your thoughts on the mzmine software. To clarify, a program was still used to assign molecular formulae to unknown peaks in an automated fashion with the constraints described in the manuscript, but it was home-built and not commercially available.

7. Generally, chapter 3.3 should describe the scan modus (e.g. full-scan with data-dependent MS2 in discovery mode).

This was clarified in the chapter and a few more details were added (line 130):

A full MS with a data-dependent MS2 acquisition was used for the present study where ions were scanned in the negative ion mode from 100 - 1500 amu (AGC target $10^6$, maximum IT 100 ms) while the top 3 most abundant ions from each scan were selected with an isolation window of 0.4 \emph{m/z} for MS/MS analysis by higher-energy collisional dissociation \citep{Olsen2007} (AGC target $5\times10^4$, maximum IT 100 ms, NCE levels stepped from 10, 30, and 50).

8. l. 165: a resolution of >100k should enable to measure four digits behind the comma with enough precision.

Having three or four digits shown when mentioning an ion in-text was inconsistent throughout, so all instances have been corrected to four digits.

9. l. 168-170: organosulfates generally ionize very efficiently in negative ESI. Hence, the peak height can be misleading with comparison to other non-sulfate-containing organic compounds.

A statement was added in the text when discussing the presence of the ion in our samples (line180):

It is noted that the relative peak intensity of this and other organosulfate species detected could be misleading when comparing to other organic ions present, as ESI ionization is known to be particularly sensitive to organosulfate compounds \citep{Surratt2008}.

10. l. 182: It should be possible to evaluate in the dataset whether isomers are present.

This is certainly possible given that we have the dimension of chromatographic retention time in our dataset, and in fact we can observe multiple peaks when presenting extracted ion chromatograms of some of the key marker ions (Figs. S5 and S7).  However, for our analysis we averaged the chromatogram (1 – 10 min) to obtain a peak list, and from that point on we lost this retention time dimension.  To clarify the referenced line, this note was simply added to point at that because the entire retention time range was averaged, what is depicted in the mass spectra is the average of all isomers for a given m/z assigned.

11. l. 269: methacryloyl peroxynitrate (two words)?

Amended.

12. l. 286: The H/C of this CHOS specie does not fit to PAHs. I did not find this formula in Jiang et al. where PAH-OS have H/C of around 1.

In Jiang et al. (2022), the formula $C_6H_{12}O_4S$ is included in the SI (Table S6) which lists compounds that were detected in both their experiments and ambient samples.  Upon closer inspection, this particular compound was only detected in systems where PAHs were mixed with DMSO, or when DMSO was being oxidized alone.

This is now clarified in the text (line 314):

For instance, the most abundant CHOS species in DEC was \ce{C6H12O4S}, \emph{m/z} 179.0385, and is attributed to the \ce{SO2}-initiated photooxidation of polycyclic aromatic hydrocarbons, such as those emitted during biomass burning \citep{Jenkins1996}, and DMSO \citep{Jiang2022}, an oxidation product of dimethylsulfide (DMS), a compound associated with marine emissions \citep{Andreae1983} as well as biomass burning \citep{Meinardi2003}.

13. l. 289-292: these two sentences are contradictory.

To clarify, monoterpene-derived organosulfates (MT-OS) were detected in low abundance compared to isoprene-derived organosulfates.  In the following sentence, I compare the number of detected MT-OS species between the different months, stating that DEC had the most.

This point is now clarified in the text (line 319):

Monoterpene-derived organosulfate species (MT-OS) were also identified, though in low abundance relative to isoprene-derived organosulfates, with the sum of MT-OS compounds in each month contributing <0.7\% to total ion signal. Interestingly, comparing the different months, the highest number of MT-OS compounds (8) were identified in DEC, including \ce{C7H12O7S}, \ce{C9H16O6S}, and \ce{C7H12O6S}.

14. l. 355-360: I find it interesting that the source of sulfate is not clear. Can this somehow be figured out based on SO2 measurements or the variability between SO2 and other marine markers -> answering whether the sulfate is coming from the ocean or from anthropogenic sources on land?

Yes, the source of sulfate remains somewhat nebulous... Given that the prevailing winds for each period studied were from the east, a marine contribution of sulfur is certainly at play to some extent, and likely a year-round influence. Specifically, methanesulfonic acid, a secondary product of DMS oxidation associated with marine environments, has been shown to contribute to particle nucleation in the atmosphere. From the HYSPLIT (Fig. 3) we see that the easterly winds do routinely pass over the city of Santarém (50 km NE of site) and also somewhat pass over the larger city of Belém (730 km east of site), so an anthropogenic component cannot be ruled out for each period as well. A potential SO2 source we did not previously mention that would have a seasonal dependence would be biomass burning, which could serve as an additional source in the dry season.

A sentence regarding a potential biomass burning influence was added in the discussion (line 309):

Besides the influence from NOx emissions, it should be noted that biomass burning can also serve as a source of SO2 and sulfate \citep{Rickly2022, Ren2021}.

Methanesulfonic acid is also mentioned now in the discussion on potential sulfate sources (line 394):

Given that easterly winds were dominant during the sampling periods (Fig.~\ref{HYSPLIT}A), a key source of sulfate is likely oxidation of dimethylsulfide (DMS) given its aforementioned association with marine environments \citep{Andreae1983}, forming methanesulfonic acid and sulfuric acid as secondary products\citep{vonGlasow2004}.

15. l. 369: I doubt that the particles nucleated over the ocean where the biogenic VOCs are missing.

To clarify, we suspect the type of particles to nucleate over the ocean may be different from those nucleated over the Amazon free troposphere. From the Atlantic we might

expect a contribution of freshly nucleated sulfate seeds upon which biogenic VOC products could condense onto as they made their way to the measurement site. Such seeds would not be as necessary in the free troposphere where temperature/surface area is low so one may expect a more purely organic oxidation product-driven nucleation could occur.

This distinction is made clearer in the discussion (line 411):

In general, the molecular volatility distributions taken together with the size distribution data suggest that the secondary UAPs measured here were likely nucleated elsewhere such as over the Atlantic ocean (e.g., sulfuric or methanesulfonic acid nucleation) or the Amazon free troposphere (e.g., organic vapor nucleation) and what was sampled was mostly aged aerosol components ranging from low to intermediate volatility.

References:

Isaacman-VanWertz, G. and Aumont, B.: Impact of organic molecular structure on the estimation of atmospherically relevant physicochemical parameters, Atmos. Chem. Phys., 21, 6541–6563, https://doi.org/10.5194/acp-21-6541-2021, 2021.

Thoma, M., Bachmeier, F., Gottwald, F. L., Simon, M., and Vogel, A. L.: Mass spectrometry-based Aerosolomics: a new approach to resolve sources, composition, and partitioning of secondary organic aerosol, Atmos. Meas. Tech., 15, 7137–7154, https://doi.org/10.5194/amt-15-7137-2022, 2022.

RC2: Anonymous Referee #2, 15 Sep 2024

Accept, minor revisions

The manuscript entitled "Seasonal Investigation of Ultrafine Particle Composition in an Eastern Amazonian Rainforest" by Thomas et al. presents an analysis of the composition of ultrafine particles (<100 nm) in the Tapajós National Forest during three seasonal periods. The authors collected size-selected particles (5-70 nm) and analysed them using high-resolution mass spectrometry. The results show that isoprene organosulfate chemistry consistently influenced ultrafine particles throughout the seasons, while biological spore fragmentation influenced particle composition in the late wet season. Biomass combustion and secondary aerosol chemistry were prominent during the dry season, leading to higher oxidation states in the particles. Organic sulfur species, due to their low volatility, are suggested to be the main drivers of new particle growth in the region.

The findings in this manuscript are certainly a valuable addition to the existing literature and my recommendation is that the manuscript should be published after some minor changes and improvements. From a scientific perspective, the manuscript clearly fits the scope of ACP. It is a thorough and interesting study, relevant for the field of atmospheric research and beyond. From a formal perspective, the quality of the manuscript is high - it is well-written, the figures and tables are clear, and all arguments and aspects are presented clearly.

We thank the reviewer for their insightful feedback and comments, which has helped improve the manuscript.

Below, you will find some general comments that the authors may want to consider:

Page 2, line 33: the term "aerosol reservoirs" seem misleading here – I suggest to look for a more appropriate word.

We agree this could be better-phrased. The referenced sentence was amended to make the meaning clearer (line 32):

Although known for its pristine air quality \citep{Andreae2004}, the Amazon basin nevertheless serves as an aerosol source of global significance thanks in part to its large area and abundant biogenic volatile organic compound (BVOC) emissions \citep{Guenther2012}, directly impacting the global CCN budget as well as helping maintain stratospheric aerosol levels \citep{Williamson2019, Weigel2011,Brock1995}.

Page 2, line 38: Please cite some more and ideally more recent studies on the absence of NPF in the Amazon here.

The following papers were added that specifically discuss the lack of NPF observations near the Amazon floor:

Wang, J., Krejci, R., Giangrande, S., Kuang, C., Barbosa, H. M. J., Brito, J., Carbone, S., Chi, X., Comstock, J., Ditas, F., Lavric, J., Manninen, H. E., Mei, F., Moran-Zuloaga, D., Pöhlker, C., Pöhlker, M. L., Saturno, J., Schmid, B., Souza, R. A. F., … Martin, S. T. (2016). Amazon boundary layer aerosol concentration sustained by vertical transport during rainfall. *Nature*, *539*(7629). https://doi.org/10.1038/nature19819

Andreae, M. O., Afchine, A., Albrecht, R., Holanda, B. A., Artaxo, P., Barbosa, H. M. J., Borrmann, S., Cecchini, M. A., Costa, A., Dollner, M., Fütterer, D., Järvinen, E., Jurkat, T., Klimach, T., Konemann, T., Knote, C., Krämer, M., Krisna, T., Machado, L. A. T., … Ziereis, H. (2018). Aerosol characteristics and particle production in the upper troposphere over the

Amazon Basin. *Atmospheric Chemistry and Physics*, *18*(2), 921–961. https://doi.org/10.5194/acp-18-921-2018

Page 6, line 176f.: Are there any indication for sesquiterpene-derived SOA in the data?

We thank the reviewer for the recommendation.  Upon consulting the literature, we were able to identify ten species bearing the same molecular formulae as known markers for sesquiterpene-derived secondary organic aerosol.  In general, these products show greater abundance in our two dry season samples, with one species in the top 20 of the most abundant assigned molecules in SEP overall: $C_{13}H_{20}O_5$, tentatively identified as an oxidation product of b- caryophyllene that was previously observed in central Amazon PM1.  We edited Figure 2 as well as Table S1 to include sesquiterpenes as an emission class category, and included the following discussion in the main text (line 250):

In addition to monoterpene markers, several sesquiterpene oxidation products were also observed (Fig.~\ref{markers}A), though in lower abundance comparatively.  The most abundant product identified had a molecular formula \ce{C13H20O5} (\emph{m/z} 255.1238), with a suggested identity of $\beta$-nocaryophyllonic acid, a multigeneration oxidation product of $\beta$-caryophyllene previously identified in Central Amazon PM$_1$ \citep{Yee2018}.  In general, observed sesquiterpene products are most abundant in SEP, contributing 1.6\% to total (assigned and unassigned) ion intensity compared to 1.2\% in DEC and JUN, consistent with the occurrence of peak abundance in observed monoterpene products.  It is notable however that these ion fractions are the most consistent between the three samples compared to all the other emission source classes, suggesting that sesquiterpene chemistry may serve as a year-round source for ultrafine particle growth in the region.

Page 4, line 89: The measurement site is west (= downwind) from the rather heavily used highway BR163. What does this mean for the measurements and data? How would signals from traffic or other human-made emissions along the highway show up in the aerosol sizing data? How can potential fossil fuel combustion be excluded as contamination for the chemical signatures? Further, how representative is this site for UFA processes in the Amazon?

This is a very important point, as highway emissions could indeed be impacting particle formation and chemistry not just at the site but the surrounding forest as well.  A potentially important impact that immediately comes to mind is the contribution of these emissions to local NOx concentrations, possibly impacting terpenoid oxidation chemistry as discussed in the manuscript.  While vehicle emissions could serve as a local source of

ultrafine particles as well, the consistency in the shape of the size distribution data suggests that this contribution is seemingly nevertheless similar between the different periods, though no definitive conclusions can be made without local traffic data. As discussed in the manuscript, higher particle concentrations in the dry season are likely driven by regional wildfire activity, consistent with seasonal observations from a site in the central Amazon (Rizzo et al. 2018). Considering the possible influence of these emissions, the authors nevertheless think that measurements made at this site are representative of the region given what is known regarding anthropogenic activity in the Eastern Amazon forest compared to other regions, including greater extents of deforestation and land use change.

The proximity of the measurement site to BR163 is made clearer in the site description (line 90):

Measurements were performed at the Santar\'{e}m-Km67-Primary Forest (BR-Sa1) tower site in the Tapaj\'{o}s National Forest (2.857° S, 54.959° W), located $\sim$7 km west of kilometer 67 of the Santar\'{e}m-Cuiab\'{a} highway and about 50 km south of Santar\'{e}m, P\'{a}ra, Brazil.

A discussion on the possible influence of highway emissions on specifically local NOx chemistry is also included (line 304):

In addition to wildfire activity, it is also likely that traffic emissions from the Santar\'{e}m-Cuiab\'{a} highway, located $\sim$7 km east of the measurement site, also contribute to local NOx concentrations. Given the dominance of easterly winds during sampling in each period (Fig.~\ref{HYSPLIT}A), local highway emissions could impact particle chemistry near the site and the larger Tapaj\'{o}s National Forest area potentially year-round, and could serve as a particularly important source of NOx in the forest during the wet season, when fire activity is low.

Page 4, line 97: how can it be excluded that multiply charged particles are also collected in the size range from 5 to 70 nm? Could this mix larger-sized particles in the ultrafine size range?

The effect of multiple charges is certainly a factor, but fortunately most of the multiply charged species collected should still have been in the ultrafine range by volume. Considering particles with the upper electrical mobility diameter limit of 70 nm as an example, and assuming (1) a Boltzmann distribution of charges as one would expect after a Po-210 bipolar neutralizer and (2) the concentration of particles larger than 70 nm is equal

to the concentration at that size (which is reasonable considering what we know about size distributions in the Amazon), we can estimate that 84% of the particles collected are singly charged, 14% are doubly charged, and 2% are triply charged, and so on. Upon calculating a volume-weighted mean diameter collected after considering these first three charge states, we find that in this scenario the average particle size collected has a diameter of 79 nm, well within the ultrafine range.

A note regarding the uncertainty in using a nano-dma for particle size selection was added (line 101):

It should be noted that particles with diameters $>$100 nm could be sampled when using a nano-DMA for size selection given what is known regarding the charge state distribution of aerosol particles after a Po-210 neutralizer \citep{Liu1986}, though this effect should be reduced by using a selected upper mobility diameter limit $<$100 nm such as 70 nm used here.

The collection of aerosol particles down to 5 nm can be experimentally challenging. How was the inlet at the tower designed and what does the transmission curve look like for the flow rate and sampling conditions chosen here?

The inlet used for this study was constructed from 3/8" copper tubing with insulation on the outside to protect it from the elements. The flows used were chosen to ensure sample flow was laminar throughout. Besides the flows contributed by the cpc and particle sampler used here, there was additionally a 10 LPM transport flow, for a total air flow of 13.5 LPM. Below is a particle penetration efficiency curve (gray dashed line) considering the dimensions of our inlet plotted with the particle-volume size distributions (dV/dlogDp) for the three months studied (solid, using the color conventions throughout the text).

[Figure]

As we can see, a significant number of particles are lost at the smallest sizes (up to 40% below 10 nm) but in terms of their contribution to particle volume sampled, these losses are negligible. Most of the particle volume collected was in fact from the highest size bins (upper limit 70 nm).

We added some additional details concerning our inlet in the main text (line 97):

Aerosol particles were sampled near the top of the tree canopy by connecting ground-level instruments to a 30~m long inlet constructed of 3/8" copper tubing and sampling at a flow of 13.5 L/min (Fig. S1 in the Supplement).

In the same section, we also noted the flow rate used by our DMA-spot sampler system (3.2 L/min) (line 101):

Particles were sampled into the analyzer at a flow of 3.2 L/min.

The aerosol number size distributions seem a bit high – I created an overplot of the size distributions of the present study and selected previous Amazonian studies (see figure below). How can the discrepancy in dN/dlogD be interpreted?

This is certainly interesting. One consideration would be that location may be important here, given that all the data used in your plot were taken from studies in central Amazonia. For reasons discussed above, one might expect the concentration of particles in the Eastern Amazon to be different from elsewhere in the rainforest, namely due to the greater influence from anthropogenic activities of land use change and increasing urbanization. More studies focusing on the size distribution of particles in this region are certainly needed to investigate this further, however.

Page 15, line 346ff: Cite also Franco et al. ACP, 22, 3469–3492, https://doi.org/10.5194/acp-22-3469-2022, 2022 here.

Thank you for the relevant reference, it was added into the discussion (line 377):

Previous studies \citep{Franco2022, Rizzo2018} have observed a similar seasonal dependence of particle concentrations in central Amazonia, with the dry season being dominated by accumulation mode ($\sim$100 - 600 in diameter) particles, possibly contributed by regional biomass burning events.